# SLAPS: Self-Supervision Improves Structure Learning for Graph Neural Networks

## Abstract

Graph neural networks (GNNs) work well when the graph structure is provided. However, this structure may not always be available in real-world applications. One solution to this problem is to infer the latent structure and then apply a GNN to the inferred graph. Unfortunately, the space of possible graph structures grows super-exponentially with the number of nodes and so the available node labels may be insufficient for learning both the structure and the GNN parameters. In this work, we propose the **S**imultaneous **L**earning of **A**djacency and GNN **P**arameters with **S**elf-supervision, or SLAPS, a method that provides more supervision for inferring a graph structure. This approach consists of training a denoising autoencoder GNN in parallel with the task-specific GNN. The autoencoder is trained to reconstruct the initial node features given noisy node features as well as a structure provided by a learnable graph generator. We explore the design space of SLAPS by comparing different graph generation and symmetrization approaches. A comprehensive experimental study demonstrates that SLAPS scales to large graphs with hundreds of thousands of nodes and outperforms several models that have been proposed to learn a task-specific graph structure on established benchmarks.

## 1 Introduction

Graph representation learning has grown rapidly and found applications in domains where data points define a graph (Chami et al., 2020; Kazemi et al., 2020). Graph neural networks (GNNs) (Scarselli et al., 2008) have been a key component to the success of the research in this area. Following the success of graph convolutional networks (GCNs) (Kipf & Welling, 2017) on semi-supervised node classification, several other GNN variants have been proposed for different prediction tasks on graphs (Hamilton et al., 2017; Veličković et al., 2018; Gilmer et al., 2017; Battaglia et al., 2018) and the power of these models has been studied theoretically (Xu et al., 2019; Sato, 2020).

GNNs take as input a set of node features and an adjacency matrix corresponding to the graph structure, and, for each node, output an embedding that captures not only the initial features of the node but also the features and embeddings of its neighbors. The performance of GNNs highly depends on the quality of the input graph structure and deteriorates substantially when the graph structure is noisy (see Zügner et al., 2018; Dai et al., 2018; Fox & Rajamanickam, 2019). The need for both node features and a clean graph structure impedes the applicability of GNNs to domains where one has access to a set of nodes and their features but not to their underlying graph structure, or only has access to a noisy structure. Examples of such domains include brain signal classification (Jang et al., 2019), computer-aided diagnosis (Cosmo et al., 2020), analysis of computer programs (Johnson et al., 2020), and particle reconstruction (Qasim et al., 2019).

In this paper, we address this limitation by developing a model that learns both the GNN parameters as well as an adjacency matrix simultaneously. Since the number of possible graph structures grows super-exponentially with the number of nodes (Stanley, 1973) and obtaining node labels is typically costly, the number of available labels may not be enough for learning both the GNN parameters and an adjacency matrix–especially for semi-supervised node classification. Our main contribution is to supplement the classification task with a self-supervised task that helps learn a high-quality adjacency matrix. Our self-supervision approach masks some input features (or adds noise to them) and trains a separate GNN aiming at updating the adjacency matrix in such a way that it can recover

the masked (or noisy) features. Introducing this self-supervision adds the inductive bias that a graph structure suitable for predicting the node features is also suitable for predicting the node labels.

We experiment with several classification datasets. For datasets with a graph structure, we only feed the node features to our model. The model operates on the node features and an adjacency that is learned simultaneously from data. We compare our model with different classes of methods: some which do not use the graph structure for predicting labels, some which use a fixed k-Nearest Neighbors (kNN) graph built based on a chosen similarity metric, and some which initialize the graph with kNN but then revise it throughout the training. We show that our model consistently outperforms these methods. We also show that the self-supervised task is key to the high performance of our model. As an additional contribution, we provide an implementation for simultaneous structure and parameter learning that scales to graphs with hundreds of thousands of nodes.

## 2 RELATED WORK

Existing methods that relate to this work can be grouped into the following categories.

**Similarity Graph:** One approach for inferring a graph structure is to select a similarity metric and set the edge weight between two nodes to be their similarity (Roweis & Saul, 2000; Tenenbaum et al., 2000). To obtain a sparse structure, one may create a kNN similarity graph, only connect pairs of nodes whose similarity surpasses some predefined threshold, or do sampling. As an example, Gidaris & Komodakis (2019) create a (fixed) kNN graph using the cosine similarity of the node features. Wang et al. (2019b) extend this idea by creating a fresh graph in each layer of the GNN based on the node embedding similarities in that layer as opposed to fixing a graph solely based on the initial features. Instead of choosing a single similarity metric, Halcrow et al. (2020) fuse several (potentially weak) measures of similarity. The quality of the predictions of these methods depends heavily on the choice of the similarity metric(s) and the value of $k$ for the kNN graph, or the threshold on similarity. Furthermore, designing an appropriate similarity metric may not be straightforward in some applications.

**Fully-connected Graph:** Another approach is to assume a fully-connected graph and employ GNN variants such as graph attention networks (Veličković et al., 2018; Zhang et al., 2018) or the transformer (Vaswani et al., 2017) which infer the graph structure via an attention mechanism, or infer the graph structure using additional information. This approach has been used in computer vision (e.g., Suhail & Sigal, 2019), natural language processing (e.g., Zhu et al., 2019), and few-shot learning (e.g., Garcia & Bruna, 2017), where there are not many nodes. The complexity of this approach, however, grows rapidly making it applicable only to small-sized graphs with a few thousand nodes and not scalable to the datasets we use in our experiments.

**Learnable Graph:** Instead of computing a similarity graph on the initial features, one may use a graph generator with learnable parameters. Li et al. (2018b) create a fully-connected graph based on a biliear similarity function with learnable parameters. A common approach is to learn to project the nodes to a latent space where node similarities correspond to edge weights. Wu et al. (2018) project the nodes to a latent space by learning weights for each of the input features. Cosmo et al. (2020) and Qasim et al. (2019) use a multi-layer perceptron for projection. Yu et al. (2020) use a GNN that projects the nodes into a latent space using the initial node features as well as an initial graph structure, aiming at providing a revised graph structure to the task-specific GNN. Franceschi et al. (2019) propose a model named *LDS* with a bi-level optimization setup for simultaneously learning the GNN parameters and a full adjacency matrix. Yang et al. (2019) update the input adjacency matrix based on the inductive bias that nodes belonging to the same class should be connected to each other and nodes belonging to different classes should be disconnected. Chen et al. (2020) propose an iterative approach that iterates over projecting the nodes to a latent space and constructing an adjacency matrix from the latent representations multiple times. In our experiments, we compare with several approaches from this category.

**Leveraging Domain Knowledge:** In applications where specific domain knowledge is available, one may leverage this to guide the model toward learning specific structures. For example, Johnson et al. (2020) leverage abstract syntax trees and regular languages in learning graph structures of Python programs that aid reasoning for downstream tasks. Jin et al. (2020b) train GNNs that are robust to adversarial attack by learning a cleaned version of the input poisoned adjacency matrix

using the domain knowledge that clean adjacency matrices are often sparse and low-rank and exhibit feature smoothness along connected nodes.

**Proposed Method:** Our model falls within the learnable graph category in which we use graph generators with learnable parameters to infer the adjacency matrix. We supplement the training with a self-supervised objective to increase the amount of supervision in learning a graph structure. The self-supervised objective is generic and can be combined with many of the models described above. It is in the same vein as the auxiliary tasks in the context of multi-task learning used in computer vision, natural language processing, and reinforcement learning (see, e.g., Jaderberg et al., 2016; Liebel & Körner, 2018; Alonso & Plank, 2016). The self-supervised task is inspired by the successful training procedures of several recent language models such as BERT (Devlin et al., 2018) and RoBERTa (Liu et al., 2019b). Similar self-supervision techniques have also been employed for GNNs (Hu et al., 2020b;c) (Jin et al., 2020a; You et al., 2020; Zhu et al., 2020). While we employ similar self-supervision techniques, our work differs from this line of work as we use self-supervision for learning a graph structure whereas the above methods use it to learn better (and, in some cases, transferable) GNN parameters. Specifically, we adopt the multi-task learning framework of You et al. (2020) with two differences: 1- we do not used shared parameters for the task-specific and self-supervised GNNs, and 2- instead of using a fixed adjacency matrix provided as input, we allow both GNNs to provide gradients for a generator that learns to generate a graph structure suitable for the downstream task.

## 3 BACKGROUND AND NOTATION

We use lowercase letters to denote scalars, bold lowercase letters to denote vectors and bold uppercase letters to denote matrices. $I$ represents an identity matrix. For a vector $v$, we represent its $i^{\text{th}}$ element as $v_i$. For a matrix $M$, we represent the $i^{\text{th}}$ row as $M_i$ and the element at the $i^{\text{th}}$ row and $j^{\text{th}}$ column as $M_{ij}$. For an attributed graph, we use $n$, $m$ and $f$ to represent the number of nodes, edges, and features respectively, and denote the graph as $\mathcal{G} = \{\mathcal{V}, A, X\}$ where $\mathcal{V} = \{v_1, \ldots, v_n\}$ is a set of nodes, $A \in \mathbb{R}^{n \times n}$ is a (sparse) adjacency matrix with $A_{ij}$ indicating the weight of the edge from $v_i$ to $v_j$ ($A_{ij} = 0$ implies no edge), and $X \in \mathbb{R}^{n \times f}$ is a matrix whose rows correspond to node features or attributes. A degree matrix $D$ for a graph $\mathcal{G}$ is a diagonal matrix where $D_{ii} = \sum_j A_{ij}$.

Graph convolutional networks (GCNs) are a powerful variant of GNNs. For a graph $\mathcal{G} = \{\mathcal{V}, A, X\}$ with a degree matrix $D$, layer $l$ of the GCN architecture can be defined as $H^{(l)} = \sigma(\hat{A} H^{(l-1)} W^{(l)})$ where $\hat{A}$ represents a normalized adjacency matrix, $H^{(l-1)} \in \mathbb{R}^{n \times d_{l-1}}$ represents the node representations in layer $l$-$1$ with $H^{(0)} = X$, $W^{(l)} \in \mathbb{R}^{d_{l-1} \times d_l}$ is a weight matrix, $\sigma$ is an activation function such as ReLU, and $H^{(l)} \in \mathbb{R}^{n \times d_l}$ is the updated node embeddings. For undirected graphs where the adjacency is symmetric, $\hat{A} = D^{-\frac{1}{2}}(A + I)D^{-\frac{1}{2}}$ corresponds to a row-and-column normalized adjacency with self-loops, and for directed graphs where the adjacency is not necessarily symmetric, $\hat{A} = D^{-1}(A + I)$ corresponds to a row normalized adjacency matrix with self-loops.

## 4 SLAPS: SIMULTANEOUS LEARNING OF ADJACENCY MATRIX AND GNN PARAMETERS WITH SELF-SUPERVISION

We break our model into four components: 1) generator, 2) adjacency processor, 3) classifier, and 4) self-supervision. The generator takes the node features as input and generates a (perhaps sparse, non-normalized, and non-symmetric) matrix $\tilde{A} \in \mathbb{R}^{n \times n}$. $\tilde{A}$ is then fed into the adjacency processor which outputs $A \in \mathbb{R}^{n \times n}$ corresponding to a normalized, and in some cases symmetric version of $\tilde{A}$. The classifier is a GNN that receives $A$ as well as the node features as input and classifies the nodes into a set of predefined classes. The self-supervision component is a GNN that receives noisy features and the generated adjacency as input and aims at denoising the features. Figure 1 illustrates the different components and in what follows, we describe each component in more detail.

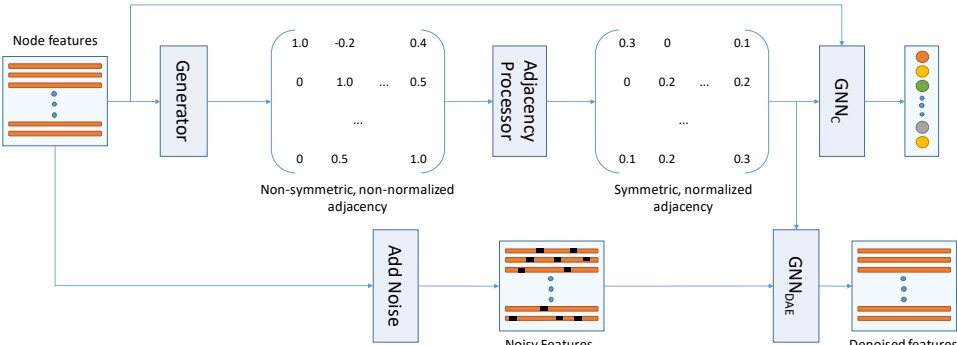

Figure 1: Overview of SLAPS. At the top, a generator receives the node features and produces a non-symmetric, non-normalized adjacency having (potentially) both positive and negative values (Section 4.1). The adjacency processor makes the values positive, symmetrizes and normalizes the adjacency (Section 4.2). The resulting adjacency and the node features go into $\mathsf{GNN_C}$ which predicts the node classes (Section 4.3). At the bottom, some noise is added to the node features. The resulting noisy features and the generated adjacency go into $\mathsf{GNN_{DAE}}$ which then denoises the features (Section 4.4).

## 4.1 GENERATOR

The generator is a function $\mathsf{G} : \mathbb{R}^{n \times f} \to \mathbb{R}^{n \times n}$ with parameters $\theta_\mathsf{G}$ which takes the node features $\boldsymbol{X}$ as input and produces $\tilde{\boldsymbol{A}} \in \mathbb{R}^{n \times n}$ as output. We consider the following two generators.

**Full Parameterization (FP):** In this case, $\theta_\mathsf{G}$ is a single matrix in $\mathbb{R}^{n \times n}$ and the generator function is defined as $\tilde{\boldsymbol{A}} = \mathsf{G}_{FP}(\boldsymbol{X}; \theta_\mathsf{G}) = \theta_\mathsf{G}$. That is, the generator ignores the input node features and directly optimizes the adjacency matrix. The disadvantages of this generator include adding $n^2$ parameters to the model, which limits scalability and makes the model susceptible to overfitting, and not being applicable to inductive settings where during test time predictions are to be made for nodes unseen during training. This generator is similar to the one proposed by Franceschi et al. (2019) except that they treat each element of $\tilde{\boldsymbol{A}}$ as the parameter of a Bernoulli distribution and sample graph structures from these Bernoulli distributions.

**MLP-kNN:** In this case, $\theta_\mathsf{G}$ corresponds to the weights of a multi-layer perceptron (MLP) and $\tilde{\boldsymbol{A}} = \mathsf{G}_{\mathsf{MLP}}(\boldsymbol{X}; \theta_\mathsf{G}) = \mathsf{kNN}(\mathsf{MLP}(\boldsymbol{X}))$, where $\mathsf{MLP} : \mathbb{R}^{n \times f} \to \mathbb{R}^{n \times f'}$ is an MLP that produces a matrix with updated node representations $\boldsymbol{X}'$; $\mathsf{kNN} : \mathbb{R}^{n \times f'} \to \mathbb{R}^{n \times n}$ produces a sparse matrix. Let $\boldsymbol{M} \in \mathbb{R}^{n \times n}$ with $\boldsymbol{M}_{ij} = 1$ if $v_j$ is among the top $k$ similar nodes to $v_i$ and $0$ otherwise, and let $\boldsymbol{S} \in \mathbb{R}^{n \times n}$ such that $\boldsymbol{S}_{ij} = \mathsf{Sim}(\boldsymbol{X}'_i, \boldsymbol{X}'_j)$ for some differentiable similarity function $\mathsf{Sim}$ (we used cosine in our experiments). Then $\tilde{\boldsymbol{A}} = \mathsf{kNN}(\boldsymbol{X}') = \boldsymbol{M} \odot \boldsymbol{S}$ where $\odot$ represents the Hadamard (element-wise) product. Since $\boldsymbol{S}$ is computed based on $\boldsymbol{X}'$, the gradients flow to the elements in $\boldsymbol{X}'$ (and consequently to the weights of the MLP) through $\boldsymbol{S}$. In the backward phase of our model, we compute the gradients only with respect to those elements in $\boldsymbol{S}$ whose corresponding value in $\boldsymbol{M}$ is $1$ (i.e. those elements $\boldsymbol{S}_{ij}$ such that $\boldsymbol{M}_{ij} = 1$); the gradient with respect to the other elements is $0$. With this formulation, in the forward phase of the network, one can first compute the matrix $\boldsymbol{M}$ using an off-the-shelf k-nearest neighbors algorithm and then compute the similarities in $\boldsymbol{S}$ only for pairs of nodes where $\boldsymbol{M}_{ij} = 1$. Unlike FP, this generator can be used for the inductive setting.

**Smart initialization:** In our experiments, we found the initialization of the generator parameters (i.e. $\theta_\mathsf{G}$) to be important. Let $\boldsymbol{A}^{kNN}$ represent an adjacency matrix created by applying a kNN function on the initial node features. One smart initialization for $\theta_\mathsf{G}$ is to initialize them in a way that the generator generates $\boldsymbol{A}^{kNN}$ before training starts (i.e. $\tilde{\boldsymbol{A}} = \boldsymbol{A}^{kNN}$ before training starts). Such an initialization can be trivially done for the FP generator by initializing $\theta_\mathsf{G}$ to $\boldsymbol{A}^{kNN}$, but may not be straightforward for the MLP-kNN generator. To enable initializing the parameters of the MLP-kNN generator in a way that it generates $\boldsymbol{A}^{kNN}$ before training starts, we consider two variants of this generator. In one, hereafter referred to simply as MLP, we keep the input dimension the same throughout the layers. In the other, hereafter referred to as MLP-D, we consider MLPs with diagonal

weight matrices (i.e., except the main diagonal, all other parameters in the weight matrices are zero). For both variants, we initialize the weight matrices in $\theta_{\mathsf{G}}$ with the identity matrix to ensure that the output of the MLP is initially the same as its input and the kNN graph created on these outputs is equivalent to $\boldsymbol{A}^{kNN}$. MLP-D can be thought of as assigning different weights to different features and then computing node similarities. Note that, alternatively, one may use other MLP variants but pre-train the weights to output $\boldsymbol{A}^{kNN}$ before the main training starts.

## 4.2 Adjacency Processor

The output $\tilde{\boldsymbol{A}}$ of the generator may have both positive and negative values, may be non-symmetric and non-normalized. To ensure all values of the adjacency are positive and make the adjacency symmetric and normalized, we apply the following function to $\tilde{\boldsymbol{A}}$:

$$\boldsymbol{A} = \boldsymbol{D}^{-\frac{1}{2}}\Big(\frac{\mathsf{P}(\tilde{\boldsymbol{A}}) + \mathsf{P}(\tilde{\boldsymbol{A}})^T}{2}\Big)\boldsymbol{D}^{-\frac{1}{2}} \tag{1}$$

Here $\mathsf{P}$ is a function with a non-negative range. In our experiments, when using an MLP generator, we apply the ReLU function to the elements of $\tilde{\boldsymbol{A}}$. When using the fully-parameterized (FP) generator, applying ReLU results in a gradient flow problem as any edge whose corresponding value in $\tilde{\boldsymbol{A}}$ becomes less than or equal to zero stops receiving gradient updates. For this reason, for FP we apply the ELU function to the elements of $\tilde{\boldsymbol{A}}$ and then add a value of 1. The sub-expression $\frac{\mathsf{P}(\tilde{\boldsymbol{A}}) + \mathsf{P}(\tilde{\boldsymbol{A}})^T}{2}$ makes the resulting matrix $\mathsf{P}(\tilde{\boldsymbol{A}})$ symmetric. To understand the reason for taking the mean of $\mathsf{P}(\tilde{\boldsymbol{A}})$ and $\mathsf{P}(\tilde{\boldsymbol{A}})^T$, assume $\tilde{\boldsymbol{A}}$ is generated by $\mathsf{G}_{\mathsf{MLP}}$. If $v_j$ is among the $k$ most similar nodes to $v_i$ and vice versa, then the strength of the connection between $v_i$ and $v_j$ will remain the same. However, if, say, $v_j$ is among the $k$ most similar nodes to $v_i$ but $v_i$ is not among the top k for $v_j$, then taking the average of the similarities reduces the strength of the connection between $v_i$ and $v_j$. Finally, once we have a symmetric adjacency with non-negative values, we compute the degree matrix $\boldsymbol{D}$ for $\frac{\mathsf{P}(\tilde{\boldsymbol{A}}) + \mathsf{P}(\tilde{\boldsymbol{A}})^T}{2}$ and normalize $\frac{\mathsf{P}(\tilde{\boldsymbol{A}}) + \mathsf{P}(\tilde{\boldsymbol{A}})^T}{2}$ by multiplying it left and right with $\boldsymbol{D}^{-\frac{1}{2}}$.

## 4.3 Classifier

The classifier is a function $\mathsf{GNN}_{\mathsf{C}} : \mathbb{R}^{n \times f} \times \mathbb{R}^{n \times n} \to \mathbb{R}^{n \times |\mathcal{C}|}$ with parameters $\theta_{\mathsf{GNN}_{\mathsf{C}}}$. It takes the node features $\boldsymbol{X}$ and the generated adjacency $\boldsymbol{A}$ as input and provides for each node the logits for each class. $\mathcal{C}$ corresponds to the classes and $|\mathcal{C}|$ corresponds to the number of classes. We use a two-layer GCN for which $\theta_{\mathsf{GNN}_{\mathsf{C}}} = \{\boldsymbol{W}^{(1)}, \boldsymbol{W}^{(2)}\}$ and define our classifier as $\mathsf{GNN}_{\mathsf{C}}(\boldsymbol{A}, \boldsymbol{X}; \theta_{\mathsf{GNN}_{\mathsf{C}}}) = \boldsymbol{A}\mathsf{ReLU}(\boldsymbol{A}\boldsymbol{X}\boldsymbol{W}^{(1)})\boldsymbol{W}^{(2)}$ but other GNN variants can be used as well (recall that $\boldsymbol{A}$ is normalized). The training loss $\mathcal{L}_C$ for the classification task is computed by taking the softmax of the logits to produce a probability distribution for each node and then computing the cross-entropy loss.

## 4.4 Adding Self-Supervision

As explained in Section 1, in many domains, the number of labeled nodes may be insufficient for learning both the structure and the GNN parameters from data. To increase the amount of supervision for learning the structure, we propose a self-supervised approach based on denoising autoencoders (Vincent et al., 2008). Let $\mathsf{GNN}_{\mathsf{DAE}} : \mathbb{R}^{n \times f} \times \mathbb{R}^{n \times n} \to \mathbb{R}^{n \times f}$ be a GNN that takes node features as well as a normalized adjacency produced by a generator as input and provides updated node features with the same dimension as input. We train $\mathsf{GNN}_{\mathsf{DAE}}$ such that it receives a noisy version $\tilde{\boldsymbol{X}}$ of the features $\boldsymbol{X}$ as input and produces the denoised features $\boldsymbol{X}$ as output. Let $idx$ represent the indices corresponding to the elements of $\boldsymbol{X}$ to which we have added noise, and $\boldsymbol{X}_{idx}$ represent the values at these indices. The aim of the training procedure is to minimize:

$$\mathcal{L}_{DAE} = \mathsf{L}(\boldsymbol{X}_{idx}, \mathsf{GNN}_{\mathsf{DAE}}(\tilde{\boldsymbol{X}}, \boldsymbol{A}; \theta_{\mathsf{GNN}_{\mathsf{DAE}}})_{idx}) \tag{2}$$

where $\boldsymbol{A}$ is the generated adjacency matrix and $\mathsf{L}$ is a loss function. To add noise to the input features for datasets where features consist of binary vectors, in each iteration, $idx$ consists of $r$ percent of the indices of $\boldsymbol{X}$ whose values are ones and $r\eta$ percent of the indices whose values are zeros, both selected uniformly at random in each epoch. Both $r$ and $\eta$ (corresponding to the negative ratio) are hyperparameters. In this case, we add noise by setting the ones in the selected mask to zeros and $\mathsf{L}$

is the binary cross-entropy loss. For datasets where the input features are continuous numbers, $idx$ consists of $r$ percent of the indices of $\boldsymbol{X}$ selected uniformly at random in each epoch. We add noise by either replacing the values at $idx$ with zeros or by adding independent Gaussian noises to each of the features. In this case, L is the mean-squared error loss.

To understand the cruciality of the proposed self-supervision, let us consider a scenario for training the model in Figure 1 (or a model with a similar architecture) but without the self-supervised task. As training proceeds, assume that two unlabeled nodes $v_i$ and $v_j$ are not directly connected to any labeled nodes. Then, since a two-layer GCN makes predictions for the nodes based on their two-hop neighbors, the edge between $v_i$ and $v_j$ receives no supervision[1]. Figure 2 provides an example of such a scenario.

As a quantitative example, in the original structures and train/validation/test splits of Cora and Citeseer, $80.4\%$ and $89.9\%$ of the nodes, and consequently $64.6\%$ and $80.8\%$ of pairs of nodes, are not connected to any labeled/train nodes. If no supervision is provided for some edges in the graph, after training the existence and weights of these edges may end up being set randomly (or the same as their initialization value), which may be problematic during testing. With the self-supervised task, however, although these edges may not receive supervision from the main task (i.e. from GCN$_\mathsf{C}$), the supervision provided by the self-supervised task (i.e. from GCN$_\mathsf{DAE}$) helps learn an appropriate weight for them.

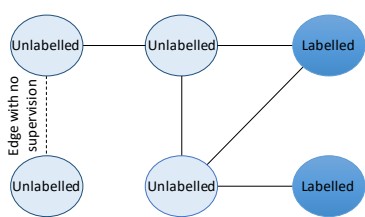

Figure 2: The dashed edge receives no supervision when training a two-layer GCN as it is not in the two-hop neighborhood of any labeled node.

### 4.5 SLAPS

Our final model, dubbed *SLAPS*, is trained to minimize $\mathcal{L} = \mathcal{L}_C + \lambda\mathcal{L}_{DAE}$ where $\mathcal{L}_C$ is the classification loss, $\mathcal{L}_{DAE}$ is the denoising autoencoder loss (see Equation 2), and $\lambda$ is a hyperparameter controlling the relative importance of the two losses.

To verify the merit of the GNN$_\mathsf{DAE}$ for learning an adjacency matrix in isolation, we also consider a variant of SLAPS named $SLAPS_{2s}$ that is trained in two stages. We first train the GNN$_\mathsf{DAE}$ model by minimizing the loss function described in Equation 2. Note that the loss function in Equation 2 depends on the parameters $\theta_\mathsf{G}$ of the generator and the parameters $\theta_{\mathsf{GNN_{DAE}}}$ of the denoising autoencoder. After every $t$ epochs of training, we fix the adjacency matrix, train a classifier with the fixed adjacency matrix, and measure classification accuracy on the validation set. We select the epoch that produces the adjacency providing the best validation accuracy for the classifier. Note that in $SLAPS_{2s}$, the adjacency matrix is trained only based on GNN$_\mathsf{DAE}$.

## 5 EXPERIMENTS

**Baselines:** We compare our proposal to several baselines with different properties. The first baseline is a multi-layer perceptron (MLP) which does not take the graph structure into account. We also compare against MLP-GAM* (Stretcu et al., 2019) which learns a fully-connected graph structure and uses this structure to supplement the loss function of the MLP toward predicting similar labels for neighboring nodes. Similar to Franceschi et al. (2019), we also consider a baseline named *kNN-GCN* where we create a kNN graph based on the node features and feed this graph to a GCN. The graph structure remains fixed in this approach. We also compare with baselines that learn the graph structure from data including LDS (Franceschi et al., 2019), GRCN (Yu et al., 2020), DGCNN (Wang et al., 2019b), and IDGL (Chen et al., 2020). We feed a kNN graph to the models requiring an initial graph structure.

**Datasets:** We use three established benchmarks in the GNN literature namely Cora, Citeseer, and Pubmed (Sen et al., 2008) as well as a newly released dataset for node classification named *ogbn-*

---

[1]While this problem may be alleviated to some extent by increasing the number of layers of the GCN, deeper GCNs typically provide inferior results due to issues such as oversmoothing (see, e.g., Li et al., 2018a; Oono & Suzuki, 2020).

Table 1: Results of SLAPS and the baselines on established node classification benchmarks. † indicates results have been taken from Franceschi et al. (2019). ‡ indicates results have been taken from Stretcu et al. (2019). Bold and underlined values indicate best and second-best mean performances respectively. OOM indicates out of memory.

| Model | Generator | Cora | Citeseer | Cora390 | Citeseer370 | Pubmed | ogbn-arxiv |
|---|---|---|---|---|---|---|---|
| *MLP* | | $56.1 \pm 1.6^{\dagger}$ | $56.7 \pm 1.7^{\dagger}$ | $65.8 \pm 0.4$ | $67.1 \pm 0.5$ | $71.4 \pm 0.0$ | $\underline{54.7 \pm 0.1}$ |
| *MLP-GAM\** | | $70.7^{\ddagger}$ | $70.3^{\ddagger}$ | – | – | $71.9^{\ddagger}$ | – |
| *kNN-GCN* | | $66.5 \pm 0.4^{\dagger}$ | $68.3 \pm 1.3^{\dagger}$ | $72.5 \pm 0.5$ | $71.8 \pm 0.8$ | $70.4 \pm 0.4$ | $49.1 \pm 0.3$ |
| *LDS* | | – | – | $71.5 \pm 0.8^{\dagger}$ | $71.5 \pm 1.1^{\dagger}$ | OOM | OOM |
| *GRCN* | | $67.4 \pm 0.3$ | $67.3 \pm 0.8$ | $71.3 \pm 0.9$ | $70.9 \pm 0.7$ | $67.3 \pm 0.3$ | OOM |
| *DGCNN* | | $56.5 \pm 1.2$ | $55.1 \pm 1.4$ | $67.3 \pm 0.7$ | $66.6 \pm 0.8$ | $70.1 \pm 1.3$ | OOM |
| *IDGL* | | $70.9 \pm 0.6$ | $68.2 \pm 0.6$ | $73.4 \pm 0.5$ | $72.7 \pm 0.4$ | $72.3 \pm 0.4$ | OOM |
| *SLAPS* | FP | $72.4 \pm 0.4$ | $\underline{70.7 \pm 0.4}$ | $\mathbf{76.6 \pm 0.4}$ | $\underline{73.1 \pm 0.6}$ | OOM | OOM |
| *SLAPS* | MLP | $72.8 \pm 0.8$ | $70.5 \pm 1.1$ | $\underline{75.3 \pm 1.0}$ | $73.0 \pm 0.9$ | $\mathbf{74.4 \pm 0.6}$ | $\mathbf{56.6 \pm 0.1}$ |
| *SLAPS* | MLP-D | $\mathbf{73.4 \pm 0.3}$ | $\mathbf{72.6 \pm 0.6}$ | $75.1 \pm 0.5$ | $\mathbf{73.9 \pm 0.4}$ | $\underline{73.1 \pm 0.7}$ | $52.9 \pm 0.1$ |

*arxiv* (Hu et al., 2020a) that is orders of magnitude larger than the other three datasets and is more challenging due to the more realistic split of the data into train, validation, and test sets. For all the datasets, we only feed the node features to the models and not the graph structure. Following Franceschi et al. (2019), we also experiment with several classification (non-graph) datasets available in scikit-learn (Pedregosa et al., 2011) including Wine, Cancer, Digits, and 20News. Dataset statistics can be found in the Appendix. For Cora and Citeseer, the LDS model uses the train data for learning the parameters of their classification GCN, half of the validation for learning the parameters of the adjacency matrix (in their bi-level optimization setup, these are considered as hyperparameters), and the other half of the validation set for early stopping and tuning the other hyperparameters. Besides experimenting with the original setups of these two datasets, we also consider a setup that is closer (although not identical) to that of LDS: we use the train set and half of the validation set for training and the other half of validation for early stopping and hyperparameter tuning. We name the modified versions Cora390 and Citeseer370 respectively where the number proceeding the dataset name corresponds to the number of labels used for training. We also follow a similar procedure for the scikit-learn datasets.

## 5.1 COMPARATIVE RESULTS

The results of SLAPS and the baselines on the node classification benchmarks are reported in Table 1. Considering only the baselines first, we see that kNN-GCN significantly outperforms MLP on Cora and Citeseer but underperforms on Pubmed and ogbn-arxiv. This shows the importance of the similarity metric and the graph structure that is fed into GCN as a low-quality structure can harm model performance. LDS outperforms MLP but the fully parameterized adjacency matrix of LDS results in memory issues for Pubmed and ogbn-arxiv. As for GRCN, it was shown in the original paper that GRCN can revise a good initial adjacency matrix and provide a substantial boost in performance. However, as evidenced by the results, if the initial graph structure is somewhat poor, GRCN's performance becomes on-par with kNN-GCN. IDGL is the best performing baseline but the iterative nature of it makes it slow to train and test.

SLAPS consistently outperforms the baselines on all datasets, in some cases by large margins. Among the generators, the winner is dataset-dependent with MLP-D mostly outperforming MLP on datasets with many features and MLP outperforming on datasets with small numbers of features. Using the software that was publicly released by the authors, all baselines that learn a graph structure fail on ogbn-arxiv and our implementation is the first that generalizes to such large graphs.

Table 2 reports the results for the scikit-learn datasets and compares with LDS and IDGL. On three out of four datasets, SLAPS outperforms the other two baselines. Among the datasets on which we can train SLAPS with the FP generator, 20news has the largest number of nodes. On this dataset, we observed that an FP generator suffers from overfitting and produces weaker results compared to other generators due to its large number of parameters.

Table 2: Results on classification datasets. † indicates results have been taken from Franceschi et al. (2019). Bold and underlined values indicate best and second-best mean performances respectively.

| Model | Generator | Wine | Cancer | Digits | 20news |
|-------|-----------|------|--------|--------|--------|
| $LDS$ | | $\mathbf{97.3 \pm 0.4}^{\dagger}$ | $94.4 \pm 1.9^{\dagger}$ | $92.5 \pm 0.7^{\dagger}$ | $46.4 \pm 1.6^{\dagger}$ |
| $IDGL$ | | $\underline{97.0 \pm 0.7}$ | $94.2 \pm 2.3$ | $92.5 \pm 1.3$ | $48.5 \pm 0.6$ |
| $SLAPS$ | FP | $96.6 \pm 0.4$ | $94.6 \pm 0.3$ | $\mathbf{94.4 \pm 0.7}$ | $44.4 \pm 0.8$ |
| $SLAPS$ | MLP | $96.3 \pm 1.0$ | $96.0 \pm 0.8$ | $92.4 \pm 0.6$ | $\mathbf{50.4 \pm 0.7}$ |
| $SLAPS$ | MLP-D | $96.5 \pm 0.8$ | $\mathbf{96.6 \pm 0.2}$ | $\underline{93.2 \pm 0.6}$ | $\underline{49.8 \pm 0.9}$ |

## 5.2 FURTHER ANALYSIS

**SLAPS$_{2s}$:** To provide more insight into the value provided by the self-supervision task on the learned adjacency, we conduct experiments with SLAPS$_{2s}$. Recall from Section 4.5 that in SLAPS$_{2s}$, the adjacency is learned only based on the self-supervision task and the node labels are only used for early stopping, hyperparameter tuning, and training GCN$_C$. Figure 3(a) shows the performance of SLAPS and SLAPS$_{2s}$ on Cora and compares them with kNN-GCN. Although SLAPS$_{2s}$ does not use the node labels in learning an adjacency matrix, it outperforms kNN-GCN ($8.4\%$ improvement when using an FP generator). With an FP generator, SLAPS$_{2s}$ even achieves competitive performance with SLAPS; this is mainly because FP does not leverage the supervision provided by GCN$_C$ toward learning generalizable patterns that can be used for nodes other than those in the training set. These results corroborate the effectiveness of the self-supervision task for learning an adjacency matrix. Besides, the results show that learning the adjacency using both self-supervision and the task-specific node labels results in higher predictive accuracy.

**The value of $\lambda$:** Figure 3(b) shows the performance of SLAPS[2] on Cora and Citeseer with different values of $\lambda$. When $\lambda = 0$, corresponding to removing self-supervision, the model performance is somewhat poor. As soon as $\lambda$ becomes positive, both models see a large boost in performance showing that self-supervision is crucial to the high performance of SLAPS. Increasing $\lambda$ further provides larger boosts until it becomes so large that the self-supervision loss dominates the classification loss and the performance deteriorates. Note that with $\lambda = 0$, SLAPS with the MLP generator becomes a variant of the model proposed by Cosmo et al. (2020), but with a different similarity function.

**Importance of $k$ in kNN:** Figure 3(c) shows the performance of SLAPS on Cora for three graph generators as a function of $k$ in kNN. For all three cases, the value of $k$ plays a major role in model performance. The FP generator is the least sensitive because in FP, $k$ only affects the initialization of the adjacency matrix but then the model can change the number of neighbors of each node. For MLP and MLP-D, however, the number of neighbors of each node remains close to $k$ (but not necessarily equal as the adjacency processor can add or remove some edges) and the two generators become more sensitive to $k$. For larger values of $k$, the extra flexibility of the MLP generator enables removing some of the unwanted edges through the function P or reducing the weights of the unwanted edges resulting in MLP being less sensitive to large values of $k$ compared to MLP-D.

**Symmetrization:** To symmetrize the adjacency, in Equation 1 we took the average of $P(\tilde{A})$ and $P(\tilde{A})^T$. Here we also consider two other choices: 1) $\max(P(\tilde{A}), P(\tilde{A})^T)$, and 2) not symmetrizing the adjacency (i.e. using $P(\tilde{A})$). Figure 3(d) compares these three choices on Cora and Citeseer with an MLP generator (other generators produced similar results). On both datasets, symmetrizing the adjacency provides a performance boost. Compared to mean symmetrization, max symmetrization performs slightly worse. This may be because max symmetrization does not distinguish between the case where both $v_i$ and $v_j$ are among the $k$ most similar nodes of each other and the case where only one of them is among the $k$ most similar nodes of the other.

**Analysing the learned adjacency:** Many graph-based semi-supervised classification models are based on the *cluster assumption* according to which nearby nodes are more likely to share the same label (Chapelle & Zien, 2005). To verify the quality of the adjacency matrix learned using SLAPS, for every pair of nodes in the test set, we compute the odds of the two nodes sharing the same label as a function of the normalized weight of the edge connecting them. Figure 3(e) represents the odds

---

[2]The generator used in this experiment is MLP; other generators produced similar results.

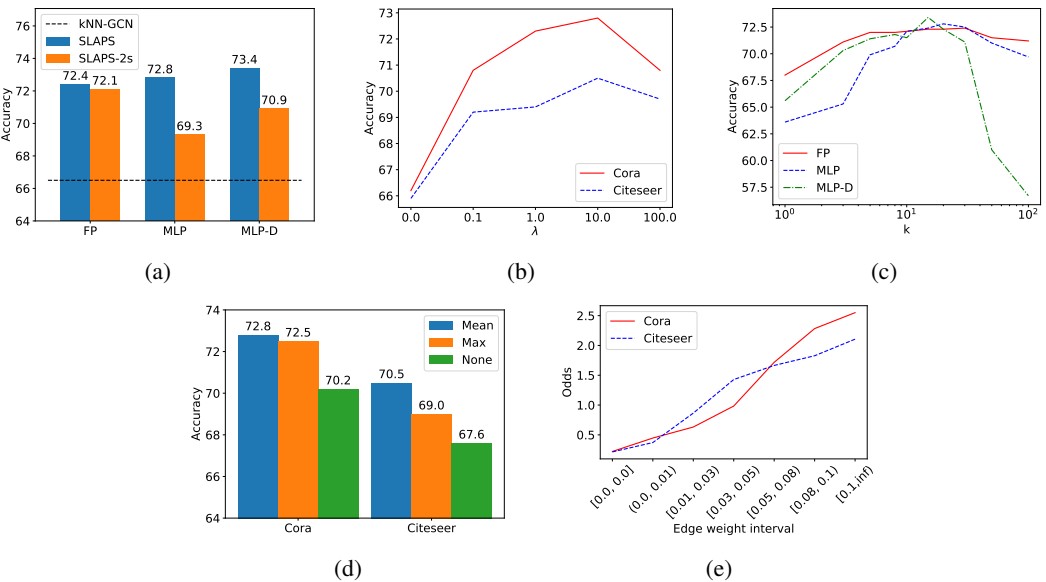

Figure 3: The performance of SLAPS (a) compared to SLAPS$_{2s}$ on Cora with different generators, (b) with MLP graph generator on Cora and Citeseer as a function of $\lambda$, (c) with different graph generators on Cora as a function of $k$ in kNN, and (d) on Cora and Citeseer with different adjacency symmetrizations. (e) The odds of two nodes in the test set sharing the same label as a function of the edge weights learned by SLAPS.

for different weight intervals. For both Cora and Citeseer, nodes connected with higher edge weights are more likely to share the same label compared to nodes with lower or zero edge weights. As a specific example, when $\boldsymbol{A}_{ij} \geq 0.1$, $v_i$ and $v_j$ are almost 2.5 times more likely to share the same label on Cora and almost 2.0 times more likely on Citeseer. Note that SLAPS may connect nodes based on a different criterion than the one used in the original datasets and so the learned adjacencies do not necessarily resemble the original structures.

## 6 CONCLUSION

In this paper, we proposed a model for learning the parameters of a graph neural network and the graph structure of the nodes simultaneously. We showed the effectiveness of our model using a comprehensive set of experiments and analyses. In the future, we would like to try more sophisticated graph generation models (e.g., GraphRNN (You et al., 2018) and GNF (Liu et al., 2019a)) and extend our approach to applications with a temporal aspect where node features are observed over time but their connections are not provided as input.

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

## A  APPENDIX

**Implementation Detail:** We implemented our model in PyTorch (Paszke et al., 2017), used deep graph library (DGL) (Wang et al., 2019a) for the sparse operations, and used Adam (Kingma & Ba, 2014) as the optimizer. We performed early stopping and hyperparameter tuning based on the accuracy on the validation set for all datasets except Wine and Cancer. For these two datasets, validation accuracy reached 100 percent with many hyperparameter settings, making it difficult to select

the best set of hyperparameters so instead, we used the validation cross-entropy loss. We fixed the maximum number of epochs to 2000. We use two-layer GCNs for both $GNN_C$ and $GNN_{DAE}$ as well as for baselines and two-layer MLPs throughout the paper (for experiments on ogbn-arxiv, although the original paper uses models with three layers and with batch normalization after each layer, to be consistent with our other experiments we used two layers and removed the normalization). We used two learning rates, one for $GCN_C$ and one for the other parameters of the models. We tuned the two learning rates from the set $\{0.01, 0.001\}$. We added dropout layers with dropout probabilities of $0.5$ after the first layer of the GNNs. We also added dropout to the adjacency matrix and tuned the values from the set $\{0.25, 0.5\}$. We set the hidden dimension of $GNN_C$ to 32 for all datasets except for ogbn-arxiv for which we set it to 256. We used cosine similarity for building the kNN graphs and tuned the value of $k$ from the set $\{15, 20, 30\}$. We tuned $\lambda$ ($\lambda$ controls the relative importance of the two losses) from the set $\{0.1, 1, 10, 100\}$. The code of our experiments will be available upon acceptance of the paper.

For GRCN (Yu et al., 2020), DGCNN (Wang et al., 2019b), and IDGL (Chen et al., 2020), we used the code released by the authors and tuned the hyperparameters as suggested in the original papers. The results of LDS (Franceschi et al., 2019) are directly taken from the original paper. All the results for our model and the baselines are averaged over 10 runs. We report the mean and standard deviation.

**Dataset statistics:** The statistics of the datasets used in the experiments can be found in Table 3.

Table 3: Dataset statistics.

| Dataset | Nodes | Edges | Classes | Features | Label rate |
|---------|-------|-------|---------|----------|------------|
| Cora | 2,708 | 5,429 | 7 | 1,433 | 0.052 |
| Citeseer | 3,327 | 4,732 | 6 | 3,703 | 0.036 |
| Wine | 178 | 0 | 3 | 13 | 0.112 |
| Cancer | 569 | 0 | 2 | 30 | 0.035 |
| Digits | 1,797 | 0 | 10 | 64 | 0.056 |
| 20news | 9,607 | 0 | 10 | 236 | 0.021 |
| Pubmed | 19,717 | 44,338 | 3 | 500 | 0.003 |
| ogbn-arxiv | 169,343 | 1,166,243 | 40 | 128 | 0.537 |

