# OpenReview forum: "SLAPS: Self-Supervision Improves Structure Learning for Graph Neural Networks"
_ICLR.cc/2021/Conference — Reject_

### Official Review · AnonReviewer1 · 2020-10-14
**Important problem; contributions are of limited significance**

**Rating:** 5
**Confidence:** 4

**Review:**

Graph neural networks (GNNs) have become de facto methods for integrating the input graph structure and node features to learn effective node representations. However, in some domains (such as brain signals, particle reconstruction, etc.), there is access to only node features (but not the underlying graph structure). Motivated by the fact that GNNs tend to perform poorly in the absence of the graph structure, the paper
1) proposes a self-supervised framework for generating the graph structure,
2) explores the design space of the framework by comparing different graph generation methods,
3) demonstrates the framework's effectiveness on node classification datasets.



##  Pros
+ [Motivation] A strength of the paper is the motivation of the problem. It is well-known that the performance of GNNs is highly sensitive to the quality of the input graph structure. Hence, in domains such as brain signals, particle reconstruction, etc., it is necessary to generate a high-quality graph structure so that GNNs could be effectively leveraged.
+ [Presentation] The high-level ideas of the paper are easy to read with clear figures and notation.
+ [Relevance] The topic of GNNs has gained increasing attention recently such that a significant portion of the ICLR community should be interested.



## Cons
The main weaknesses of the paper are along the axis of the significance of the contributions. The paper requires thorough discussions / positioning / comparisons with many existing publications in GNN literature. The detailed comments are as follows.
- [Self Supervision] The framework proposed in the paper can be seen as an instance of a general framework for self-supervision proposed in SS-GCN (When Does Self-Supervision Help Graph Convolutional Networks?, In ICML'20).
- [Self Training] A closely related idea for self-supervision is self-training. The basic idea of self-training is to add high-confident predictions to the training set to increase supervision. The paper should be positioned with (and empirically compare against) self-training-based approaches such as (but not limited to)
i) GAM (Graph Agreement Models for Semi-Supervised Learning, In NeurIPS'19) that can also handle noisy / learn graph structures,
ii) AdaEdge (Measuring and Relieving the Over-smoothing Problem for Graph Neural Networks from the Topological View, In AAAI'20).
- [Quality] Regarding experiments, the dataset domains (such as citation networks) considered in the paper are those in which the graph structure is known. So, it would be more convincing if the adjacency of the dataset was used as the initial adjacency matrix in the proposed framework. The baselines to compare against would then be models such as SS-GCN, GAM, etc.
- [Soundness] It is unclear why datasets from domains discussed in the introduction section were not considered in the experiments. These domains include, as listed in the paper, brain signal classification, computer-aided diagnosis, analysis of computer programs, and particle reconstruction.
- [Clarity] Though the high-level ideas were easy to read, the paper should clearly discuss finer details. In particular, in section 4.1 (paragraph preceding 4.2), the details of adjacency initialisation are hastily mentioned, and it feels like the discussion mixes adjacency and weight matrices up.
- [Metric Learning] An idea for learning graph structures for GNNs is to learn the underlying similarity metric (rather than use a chosen one). This idea has been used in certain domains of computer vision and natural language processing. The paper should be positioned with relevant publications (albeit for small graphs) including (but not limited to)
i) Few-Shot Learning with Graph Neural Networks, In ICLR'18
ii) Graph Neural Networks with Generated Parameters for Relation Extraction, In ACL'19

---

> ### Author Response · Authors · 2020-11-16
> **Author Response**
>
> We thank the reviewer for valuable feedback. Our revision addresses the identified issues. The changes to the manuscript have been highlighted in blue.
>
> + When Does Self-Supervision Help Graph Convolutional Networks?
>
> This reference falls under the same category as Hu et al. (2020b;c) already referenced in the last paragraph of our related work section. We added this reference in our revision.
>
> + Self-training
>
> Self-training can be seen as orthogonal to our proposal. For any of the models in Table 1 (SLAPS or the baselines), one can learn an initial model given the training labels, then make predictions about the unlabeled nodes, then take the top M nodes for which the model is most confident, then consider the predictions for these M nodes as labels and train a fresh model with the new set of labels.
>
> + It would be more convincing if the adjacency of the dataset was used as the initial adjacency matrix
>
> We are studying a different problem setting. Our paper tackles the problem of learning from a set of nodes without having access to a graph structure, which is a different and more challenging problem than learning to revise a given adjacency matrix. The setup we used is similar to LDS.
>
> However, to verify the merit of our approach when a noisy graph structure is provided as input, we conducted an experiment on Cora where we used the original graph structure of Cora plus 20% noisy edges (added uniformly at random) as the adjacency matrix. Then we ran GCN and SLAPS on Cora with these noisy adjacency matrices 10 times. GCN achieved an accuracy of 73.3 +- 0.9 and SLAPS achieved 78.7 +- 0.7. This confirms the merit of SLAPS when an initial (noisy) graph structure is provided as input.
>
> + Using datasets from domains discussed
>
> We decided to run our experiments on the standard benchmarks that are well-known and well-understood in the graph representation learning community. We could alternatively run our experiments on the datasets from the domains discussed but note that as explained in the related work section (“Leveraging domain knowledge” paragraph), each of these datasets requires design choices guided by the domain knowledge.
>
> + The details of adjacency initialisation are hastily mentioned
>
> Our revision explains the initialization in more detail (page 4, smart initialization paragraph). Thanks for pointing this out.
>
> + 1) Few-Shot Learning with Graph Neural Networks, In ICLR'18 2) Graph Neural Networks with Generated Parameters for Relation Extraction, In ACL'19
>
> Both of these references rely on fully connected graph structures that are not scalable to the datasets in our experiments. We described this category of approaches under the “Fully-connected Graph” paragraph of our related work section. We added these two references there as well.

---

> > ### Comment · AnonReviewer1 · 2020-11-16
> > **Thanks for the response**
> >
> > Thanks for sharing a revised version of the paper. I agree with the other reviewers that the scope of the paper will be limited if SLAPS does not perform well for noisy input graphs. Moreover, I still feel the paper requires positioning with and comparisons against relevant prior work.
> >
> > More specifically,
> > 1) When a graph is not provided as input, SLAPS should be compared against GAM* as discussed in
> > ```
> > Graph Agreement Models for Semi-Supervised Learning, In NeurIPS'19.
> > ```
> > The MLP + GAM* model used in that publication does not use the graph structure and is an important baseline to compare against (for citation datasets). It would add great value to the paper if non-graph datasets (such as CIFAR and SVHN) were also considered for SLAPS (please see _section 4.2: Semi-Supervised Learning Without a Graph_ in the publication for more details).
> > 2) When the given graph is noisy, then direct comparison with GCN would be somewhat unfair.  A few competitive baselines would actually be (but not limited to) GCN + GAM, GAT + GAM, etc. (please see section 4.1 in the publication).
> > 3) The method SLAPS should be positioned with the three different schemes listed in section 3.2 of the paper (especially the _multi-task learning_ scheme)
> > ```
> > When Does Self-Supervision Help Graph Convolutional Networks?, ICML'20.
> > ```
> > Please see equation 4 in the publication, SLAPS can be seen as a special case (where $X$ contains the original features, $X_{ss}$ contains the noisy features, $\Theta$ contains $GNN_c$ parameters, and $\Theta_{ss}$ contains $GNN_{DAE}$ parameters). Multi-task learning discussed in that publication is very general and makes no assumption on the self-supervised task type.

---

> > > ### Author Response · Authors · 2020-11-19
> > > **Author Response**
> > >
> > > +Comparison with GAM*
> > >
> > > We still hold the same belief that the proposals in the GAM paper and in our paper are orthogonal. We aim at improving structure learning through a better use of the node features and GAM aims at improving it through a better use of the node labels. One can easily add to SLAPS the extra terms in the loss functions of GAM (in Equations 1 and 2). One can also easily add to GAM the extra terms in the loss function of SLAPS (i.e. the self-supervision loss) to learn a better function g. That is, one can easily have a GAM+SLAPS model. Nevertheless, as recommended by the reviewer, we added the results of MLP+GAM* in our revision.
> > >
> > > +It would add great value to the paper if non-graph datasets were also considered
> > >
> > > We have reported results on four non-graph datasets (Wine, Cancer, Digits, and 20News). We selected these datasets because they have been used in previous work (LDS and IDGL) and we thought making improvements on them would make for a stronger claim.
> > >
> > > +When Does Self-Supervision Help Graph Convolutional Networks?, ICML'20.
> > >
> > > We added a note in our revision specifying the similarities and differences.

---

> > > > ### Comment · AnonReviewer1 · 2020-11-19
> > > > **Follow-up**
> > > >
> > > > Appreciate the new changes. Would like to stress that the scope/significance of the paper would be limited if SLAPS was not shown to perform well for noisy input graphs (also pointed out by other reviewers).
> > > >
> > > > Though the proposals are different, SLAPS and GAM* both have the same motivation (i.e. learning without graph structure). Based on the statistics of the non-graph datasets (Wine, Cancer, Digits, and 20News) shown in Table 3, the datasets are quite small in size (a small number of instances and features). Thus, MLP + GAM* is a relevant (and not a time-consuming) baseline in these experiments.

---

### Official Review · AnonReviewer3 · 2020-10-28
**Learning graph structures with SSL sounds interesting in general, but some clarifications of the model design are needed**

**Rating:** 5
**Confidence:** 4

**Review:**

Summary:
This paper proposes to tackle jointly learning graph structures and GNN parameters without accessing the original graph structure. Specifically, the proposed method adopts a self-supervised auxiliary task, i.e., parallel training using the supervision of node labels and a self-supervised task using de-noised auto-encoding. The latent graph structure is generated through a fully-parameterized adjacency matrix or a KNN construction subsequent to passing node features to an MLP. Experimental results and corresponding analyses demonstrate the effectiveness of the proposed model.

Pros:
1. Using self-supervision to guide graph structure learning sounds like an interesting and promising idea.
2. Most claims are supported by experimental results. Specifically, comparisons with baselines show that the proposed model can alleviate the problems of previous works such as sensitivity to the similarity metric and high-quality initial graph structure, memory problems, etc. Datasets like Wine, 20news, etc., demonstrate the model's validity for tasks without graph structure. Parameter sensitivity and other ablation studies are also conducted.
3. Overall, the paper is well written and easy to follow.

Cons:
1. Some model designs are not entirely clear. For example, based on my understanding, the KNN step is not differentiable, thus the model with MLP-kNN generator cannot be trained end-to-end. The authors may want to detail how to cope with this problem in the training process.
2. I also have a question regarding the model design regarding the self-supervised loss. If neglecting the supervised part (i.e., GNNc and its loss in Figure 1), the self-supervised part basically tries to reconstruct the original feature from the noisy features using GNN_DAE. In that case, should KNN using initial features be the optimal solution (since GCN is essentially smoothing features and these nodes have the most similar features)? If that is true, I feel that the self-supervision may actually work like a regularizer to the MLP subsequent to KNN.
3. The experiments show that the symmetrization step is very useful while lacking analysis of why it works. Could the authors provide further exploration in that aspect?
4. It seems bizarre that only LDS is adopted as the baseline in Table 2 since the codes and datasets are ready. I suggest the authors at least add IDGL, the most competitive baseline in Table 1, into Table 2.
4. There's little comparison of the learned graph structure with the original one (Figure 3.e only utilize node labels but not the original graph structure). Since the main goal of the paper is to learn GNN and graph structure simultaneously, the learned graph structure is important and should be analyzed empirically. For example, the authors may want to follow previous works to design certain metrics to compare the learned graph structure with the original one, provide some visualizations, or case studies in synthesis graphs.

Minor:
1. The complete parameter sensitivity could be added to the supplementary material. For example, I am interested to see how the sensitivity of parameter k for kNN behaves across different datasets with different scales.
2. Some related works are missing, e.g., [1-3].

[1] Adaptive Graph Convolutional Neural Networks, AAAI 2018.
[2] Topology Optimization based Graph Convolutional Network, IJCAI 2019.
[3] Graph Structure Learning for Robust Graph Neural Networks, KDD 2020.

Overall, though I like the paper in general, I believe the paper could be further improved and thus vote for weakly rejection. I am happy to increase my scores if the authors can address my above concerns.

---

> ### Author Response · Authors · 2020-11-16
> **Author Response**
>
> We thank the reviewer for valuable feedback. Our revision addresses the identified issues. The changes to the manuscript have been highlighted in blue.
>
> + The KNN step is not differentiable, thus the model with MLP-kNN generator cannot be trained end-to-end.
>
> We updated our MLP-kNN description in the revision and explained how the gradients are computed (page 4, MLP-kNN paragraph).
>
> + Should KNN using initial features be the optimal solution?
>
> Not necessarily. Here’s a toy counterexample. Suppose the dimension of initial node features is 5 and assume the node features for four of the nodes (denoted N1, N2, N3 and N4) in the dataset are as follows:
>
> N1: 1, 0, 1, 1, 1
>
> N2: 1, 0, 0, 0, 0
>
> N3: 0, 0, 1, 0, 1
>
> N4: 0, 0, 1, 1, 1
>
> ...
>
> Now assume k=2. Then kNN connects N1 to N3 and N4. However, with this structure, if we mask the first feature of N1 (by setting it to zero), the GNN_DAE (GNN with only the self-supervised task) will have a hard time recovering it. So, during training, GNN_DAE may learn to instead connect N1 to N2 and N4.
>
> As another example, assume the nodes are humans and suppose one of the nodes is a computer scientist and also a pianist. With a kNN graph with k equals (say) 5, this node may become connected to only computer scientists (or only to pianists) making the model lose a big part of their personality. But the GNN_DAE may learn to connect this node to, say, the 3 most similar computer scientists and 2 most similar pianists so the model will have a better picture about this node.
>
> + Symmetrization step
>
> Our intuition is that the symmetrization step helps for two reasons: 1) the similarity metrics we use are symmetric by definition (i.e. if v_i is similar enough to v_j to justify an edge from v_i to v_j, v_j is also similar enough to v_i to justify an edge from v_j to v_i) and although the kNN step produces a non-symmetric graph structure, it is beneficial to add these edges back to the graph. 2) Symmetrization is a good way of adding new useful edges to the graph without increasing the value of k in kNN, as increasing k may result in adding some good edges but also several bad edges to the graph.
>
> As for why mean symmetrization performs better than max symmetrization, we explain in the experiment reported in Figure 3(d) that information loss might be an important factor (page 9, last three lines in symmetrization paragraph).
>
> + I suggest the authors at least add IDGL, the most competitive baseline in Table 1, into Table 2.
>
> We added the IDGL results to Table 2 in our revision. Thanks for the suggestion.
>
> + Comparison of the learned graph structure with the original one
>
> As explained in the last sentence right before the conclusion, we believe comparing the structure learned by SLAPS to the original structure of the dataset is not meaningful. That’s because the original structure of the dataset is based on a relation R1 between the nodes, whereas SLAPS learns the structure based on a learned notion of similarity corresponding to a different relation R2. For Cora, for example, the original structure connects two nodes if one cites the other (i.e. R1 = citation); SLAPS, on the other hand, connects them if they are similar in a latent space (i.e. R2 = similarity in a latent space). These two relations are not necessarily equivalent, so comparing their overlap may not be meaningful.
>
> + Sensitivity to k for other datasets
>
> So far, we conducted the suggested experiment for the sensitivity of parameter k for kNN in Citeseer and the results are as follows (we will add more to our final version):
>
> K: 3, 5, 10, 15, 20, 30, 50, 100
>
> FP: 63.2, 65.6, 68.1, 67.9, 69.4, 70.7, 69.7, 69.3
>
> MLP: 65.4, 66.9, 68.3, 68.7, 69.5, 70.5, 70.1, 68.3
>
> MLP-D: 66.1, 67.7, 70.5, 73.1, 72.6, 71.7, 69.4, 65.0
>
> + The suggested references
>
> Thanks, we added the references in our revision.
>
> + I am happy to increase my scores if the authors can address my above concerns.
>
> We appreciate this and we are happy to address any other concerns you may have.

---

> > ### Comment · AnonReviewer3 · 2020-11-20
> > **Thanks for your responses and a follow-up question**
> >
> > I thank the authors for their detailed responses. I believe the paper has improved its quality after the revision. But I still have one follow-up question regarding the learned graph structure (previous Q5).
> >
> > As agreed by another reviewer, I think it is important to analyze the learned graph structure, since this is the main contribution of this paper (even though I acknowledge the ultimate goal of the proposed method is to improve the prediction results of GNNs). If the authors believe directly comparing the learned structures with the original structures in real datasets such as Cora is not appropriate, I suggest the authors at least giving an example using synthetic datasets where the ground-truth graph is known. This step is important to verify that the proposed method achieves good results because it can discover the real underlying graph.
> >
> > Besides, I agree with other reviewers that using noisy graphs as inputs is an important experiment that should be added to the paper (I only find the results in the authors' responses but not in the revised paper). The experimental setups also need to be explained clearly to ensure fair comparisons.

---

### Official Review · AnonReviewer2 · 2020-10-28
**Self-supervised loss for simultaneously learning structure and parameters for GNN**

**Rating:** 5
**Confidence:** 4

**Review:**

The authors propose a method for simultaneously learning the graph structure (or a graph generative model) and the parameters of a GNN for node classification. This is a topic of recent interest and highly relevant to the ICLR community.

The authors propose two different ways for the graph generator. First, a fully parameterized method that outputs a continuous (weighted) adjacency matrix. The second computes a spare k-NN graph based on the input similarities. Using the word “graph generator” or “generative model” is a bit of a misnomer here because both methods are not probabilistic but deterministic. The adjacency matrices output by these “generators” are then made into symmetric and positive matrices (i.e., adjacency matrices for positively weighted undirected graphs). Finally, in addition to the typical supervised loss, the authors also propose an unsupervised loss based on a reconstruction loss.

First, I like the idea of adding an unsupervised loss and also appreciate the experiments and the results as robust.

I have a concern about the approach though and perhaps this could be clarified in a conversation here. When you don’t use the MLP-kNN “generator” then the adjacency matrix generated is not sparse at all. But what that means is that you are using (for each node) a fully connected layer. Do you think that the advantage of your method then is in the “adjacency processor step”? When you do use the MLP-kNN, how do you compute the gradients? Is it that you only compute gradients for the edges you selected? In this case you would have lots of edges without “supervision” as you call it. Or are you also obtaining gradients for the edges not selected? In this case, creating the kNN graph is a discrete operation and it is not clear to me how to differentiate through such an operation. Of course, recent proposals have been made for differentiating through kNN but I don’t see any reference or mention of what you are doing here. Could you please elaborate on that?

Regarding the statements about LDS. First, you write that if two nodes v_i and v_j are not directly connected to any labeled nodes, then the RV between them (the edge) does not receive supervision. The statement (and the text following it) is somewhat misleading for two reasons.

First, both of the nodes have to be not directly connected. But then in the next sentence you wrote that 80 and 89% of nodes are not directly connected to a labeled node for the standard benchmark graphs. The more appropriate analysis, however, would be to count the number of pairs of nodes where both nodes are not connected to a labelled node. This should also include the validation nodes used for the outer objective in LDS. Also note that LDS doesn’t always use the standard benchmark graphs (it either constructs a k-NN graph where k is a hyperparameter of the method or initializes with a subgraph of the given graph).

Second, your statement is true for *one* sampled graph. Remember that LDS samples a set of graphs in each iteration. It can happen that, even if there is a pair of nodes where both nodes are not directly connected to a labelled node in one sampled graph, one of these nodes might be connected in a different sample. Indeed, if a sampled edge connecting either v_i or v_j directly to a labelled node leads to a reduction of the loss, the next time said edge is probably more likely to be sampled. This is not to say that self-supervision is not a good idea (I do like the idea) but I’m not sure if your statements about LDS are quite accurate.

Overall, I think that this paper strength is the proposed self-supervised loss and the experimental evaluation. It is rather weak on the methodology, its presentation, and related  discussion. There are several questions I need to hear your response to. Once these are clarified I'm open to adjust my score accordingly.

---

> ### Author Response · Authors · 2020-11-16
> **Author Response**
>
> We thank the reviewer for valuable feedback. Our revision addresses the identified issues. The changes to the manuscript have been highlighted in blue.
>
> + When you don’t use the MLP-kNN “generator” then the adjacency matrix generated is not sparse at all. But what that means is that you are using (for each node) a fully connected layer. Do you think that the advantage of your method then is in the “adjacency processor step”?
>
> In the case of a full parametrization (FP) generator, first of all, note that the adjacency matrix is weighted. So, although each node may be fully-connected to every other node, the weights of the connections are different. Secondly, note that we use an activation function (ELU+1) which can make a large number of the elements in the adjacency matrix have a near-zero (but not exactly zero) value. In that sense, the learned graph structure can be sparse.
>
> While the adjacency processor step plays a positive role in all our models (as shown in Figure 3(d)), the main advantage of the model comes from self-supervision (as shown in Figure 3(b)). To further test this empirically with a FP generator (instead of a MLP in Figure 3(b)), we set \lambda to zero to remove the effect of self-supervision and the accuracy on Cora dropped from 72.4 +- 0.4 to 65.4 +- 1.9. This confirms that the main advantage of the model comes from self-supervision and not from the adjacency preprocessor.
>
> + When you do use the MLP-kNN, how do you compute the gradients?
>
> We only computed gradients for the selected edges. We updated our MLP-kNN description in the revision and explained how the gradients are computed (page 4, MLP-kNN paragraph).
>
> + In this case you would have lots of edges without “supervision” as you call it.
>
> We would like to clarify with the following example.
> Take Figure 2 in the paper as an example and let us call the two unlabeled nodes that are not connected to any labeled nodes as the i-th and the j-th node.
>
> If we use SLAPS without the self-supervision part (with any of our generators), then no matter what the value of \tilde{A}_ij is (zero or non-zero), the value of this edge has no effect on the loss function. Therefore, it receives no supervision.
>
> On the other hand, for SLAPS with self-supervision, two cases might occur:
>
> if \tilde{A}_ij > 0 in an epoch (corresponding to the case where kNN selects the j-th node as a neighbor of the i-th node in MLP-kNN), the value of \tilde{A}_ij plays an important role in the loss function of GNN_DAE and receives gradients.
>
> if \tilde{A}_ij = 0 in an epoch (corresponding to the case where kNN does not select the j-th node as a neighbor of the i-th node in MLP-kNN), this edge does not play a role in this epoch and receives no supervision. However, the MLP parameters get updated based on the supervision from the other edges and \tilde{A}_ij may become non-negative in the next epochs.
>
> + The more appropriate analysis, however, would be to count the number of pairs of nodes where both nodes are not connected to a labeled node.
>
> The number of pairs of nodes is 64.6% for Cora and 80.8% for Citeseer. We updated this in the revision.
>
> + Second, your statement is true for one sampled graph. Remember that LDS samples a set of graphs in each iteration.
>
> We agree that the description was not quite precise. To be more precise, if the Bernoulli parameters corresponding to the connection between v_i and the labeled nodes as well as v_j and the labeled nodes is close to zero, then in almost all sampled graphs, v_i and v_j will not be connected to any labeled node and the random variable corresponding to the edge connecting v_i and v_j receives no supervision.
>
> + Revision based on the comments about LDS
>
> We agree with the reviewer that there are fine details in the LDS model that make the analysis in Figure 2 for LDS somewhat brittle. The goal of that analysis was not to disparage the LDS model, but to motivate the need for self-supervision. For this reason, our revision discusses Figure 2 in the context of SLAPS with and without supervision as opposed to discussing it in the context of LDS.
>
> + Once these are clarified I'm open to adjust my score accordingly.
>
> We appreciate this and we are happy to address any other concerns you may have.

---

### Official Review · AnonReviewer4 · 2020-10-28
**Interesting work with convincing results.**

**Rating:** 7
**Confidence:** 3

**Review:**

The paper proposes a way to use self-supervision with denoising autoencoders to improve learning of the graph structure for GNNs. The approach is compared with a number of recent approaches from the literature.

The approach addresses the highly relevant problem of learning graph structure with GNNs. The paper is well written, with clear motivation and an informative summary of similar prior works. The differences with respect to these approaches, primarily the self-supervised aspects, are clearly pointed out.

There are a few prior works covering self-supervision with GNNs that are all quite new [1-3]. It would strengthen the manuscript if these works were discussed in relation to this work. However, consider how new they all are it is not surprising that they were not mentioned.


Quality
- While the originality is perhaps not big, the work is thorough and well written. There is a substantial experimental comparison with relevant prior work and the conclusions seems to be substantiated.

Clarity
- The paper is well written and easy to understand. I would have liked a little more details and explanation on the MLP-kNN aspects.

Originality
- Self-supervision seems to be the topic of a lot of papers lately, so in that respect, this work is perhaps not the most original.

Significance
- I am not aware of similar works and the results are convincing, so I guess the approach could have reasonable influence on the field.

[1] Jin, Wei, et al. "Self-supervised learning on graphs: Deep insights and new direction." arXiv preprint arXiv:2006.10141 (2020).
[2] You, Yuning, et al. "When Does Self-Supervision Help Graph Convolutional Networks?." arXiv preprint arXiv:2006.09136 (2020).
[3] Zhu, Qikui, Bo Du, and Pingkun Yan. "Self-supervised Training of Graph Convolutional Networks." arXiv preprint arXiv:2006.02380 (2020)

---

> ### Author Response · Authors · 2020-11-16
> **Author Response**
>
> We thank the reviewer for valuable feedback. Our revision addresses the identified issues. The changes to the manuscript have been highlighted in blue.
>
> + The suggested references
>
> Thanks for pointing out these works. The three suggested references fall under the same category as Hu et al. (2020b;c) already referenced in the last paragraph of our related work section. We added these references to our revision.
>
> + I would have liked a little more details and explanation on the MLP-kNN aspects.
>
> We included more detail about MLP-kNN in our revision (page 4, MLP-kNN paragraph). If there are any other specific details or explanations you would like to see, please let us know as we would be happy to add them.
>
> + Originality
>
> We agree that self-supervision has been the subject of several GNN papers recently. However, to the best of our knowledge, none of these have used self-supervision as a way to address the issue that we highlight in Fig (2). In that sense, we believe our work is original.

---

### Official Review · AnonReviewer5 · 2020-11-06

**Rating:** 5
**Confidence:** 3

**Review:**

This paper considers the problem of nodes classification with few labeled data and missing graph structures. The proposed solution is expected to infer unobserved graph structure as well as the parameters of the classification model. The main contribution of this paper is proposing adding a denoise autoencoder layer which provides more supervision to the learning. The model compares favorably with other states of art models in several benchmark graph data sets.

Concerns:
1. The experiments assume inputs only contain node features. The solution proposed seems to be incremental under this setting as the problem is a special case of few-shot learning where metric-learning based methods including GNN and denoise autoencoder all have been studied before. See [1], [2], [3]. Further discussion of related works is necessary.

2. The proposed method seems very heuristic driven without providing further theoretical analysis. It is unclear how and why the self-supervision might help the classification learning task. Figure 2 gives a negative example of the existing method but it does not explain why the proposed method could make a difference.

3. How does the proposed model compare with regular GNN/GCN models if edges are not entirely missing? The scope of the paper will be limited if it doesn’t work well when graph data is noisy but not entirely missing.

Reference
[1] https://arxiv.org/abs/1905.01102
[2] http://www.cs.toronto.edu/~gkoch/files/msc-thesis.pdf
[3] https://arxiv.org/abs/1711.04043

---

> ### Author Response · Authors · 2020-11-16
> **Author Response**
>
> We thank the reviewer for valuable feedback. Our revision addresses the identified issues. The changes to the manuscript have been highlighted in blue.
>
> + The suggested references
>
> We thank the reviewer for the references; we updated the related work section accordingly. However, we believe the suggested references do not clash with the contribution and novelty of our work. We discuss each reference below:
>
> [1] Generating Classification Weights with GNN Denoising Autoencoders for Few-Shot Learning
>
> This work uses a fixed kNN graph based on cosine similarities and does not learn the graph structure during training. We have included a similar baseline in our experiments named kNN-GCN (see Table 1). Moreover, while this work uses a denoising auto-encoder (DAE), the purpose of the DAE is entirely different from the purpose of our DAE. The authors use a DAE to make the representation of the test nodes (which has been computed using only a few examples) more robust, whereas we use a DAE to provide more supervision for graph structure learning.
>
> [2] Siamese Neural Networks for One-shot Image Recognition
>
> We are not sure we understand how this work is related to the problem we study in this paper. The authors use siamese networks for one-shot image recognition and do not use any notion of graphs or structure learning. Kindly elaborate on how this work is related to ours.
>
> [3] Few-Shot Learning with Graph Neural Networks
>
> The proposed architecture in this work falls under the “Fully-connected Graph” category in our related work section. This architecture can be viewed as applying a variant of the graph attention network on a fully-connected graph. Our experiments include graphs with hundreds of thousands of nodes and using fully connected graphs is not feasible here.
>
> + Why the proposed method could make a difference
>
> According to Figure (2), without using our proposed self-supervised task, many edges receive no supervision during training. Therefore, the existence and the weight of the edge between these two nodes will be set randomly (or as the same as the initial value).
> Our self-supervised task makes sure that every edge receives supervision. That is, although the edge between the two nodes marked in Figure 2 receives no supervision from the main task, it does receive supervision from the self-supervised task. This helps learn reasonable edge weights for all possible edges in the graph, and only for those that are close to the labeled nodes. We added a note about this in our revision (see the paragraph right before Section 4.5).
>
> + How does the proposed model compare with regular GNN/GCN models if edges are not entirely missing?
>
> To verify the merit of our approach in the proposed setting, we conducted an experiment on Cora where we used the original graph structure of Cora plus 20% noisy edges (added uniformly at random) as adjacency matrix. We ran GCN and SLAPS on Cora with these noisy adjacency matrices 10 times. GCN achieved an accuracy of 73.3 +- 0.9 while SLAPS achieved 78.7 +- 0.7. This confirms the merit of SLAPS when the edges are not entirely missing.

---

> > ### Comment · AnonReviewer5 · 2020-11-17
> > **Still have concerns about this paper**
> >
> > I appreciate the efforts authors have put into the revision. I have read the other reviews and comments carefully. However, the response does not fully resolve all of my concerns as follows.
> >
> > 1. The title claims the method improves graph structure learning. Nonetheless,  the structure of the learned graph remains unexplored throughout the paper. Most experiments are for node classification tasks and under the setting without an underlying graph. Figure 3 (e) gives an illustration of the positive correlation between the learned edge weights and labels but it does not reveal much about the changes or even improvement to the structure of the graph.
> >
> > 2. I also expect the explanation of the effects of self-supervision to be more principal.  All edges in the generated graph are affected by the updates of the MLP simultaneously and hence they are claimed to be able to receive supervision from the denoising tasks. But, how exactly does this process, as a core contribution of this paper, increase the generalization ability of the GCN model? For LDS, we can know from its bilevel objective function which explicitly optimizes for generalization. For SLAPS, it is not that clear. The original denoising autoencoder paper includes understanding from multiple perspectives including Manifold learning. Do any of those relate to the proposed method? If not, what else could help interpret the experimental results?
> >
> > 3. As mentioned above, the paper has little focus on the structures of the learned graph, It is unclear whether this paper is for semi-supervised/few-shot classification problems with a graphical approach or it is for learning the graph structure based on incomplete information. If we look through the lens of the classification problems, non-graphical semi-supervised/few-shots learning models and baselines should be considered and discussed. The references listed are to reflect that the positioning of this paper needs further elaboration under this particular context.

---

> > > ### Author Response · Authors · 2020-11-18
> > > **Author Response**
> > >
> > > 1.
> > > We try to learn a graph structure that allows us to maximize classification accuracy. This is consistent with our title as our title claims we improve structure learning for GNNs, not structure learning in general. In addition to the classification results, we show in Figure 3(e) that there is a positive correlation between nodes sharing the same labels and the edge weight computed by SLAPS. However, it is true that it is difficult to interpret the learned graph structure apart from the fact that it is designed to accurately predict node labels. In this sense, the learned structure might differ significantly from the existing underlying graph. When we claim that the method improves structure learning for GNNs, we mainly refer to what we describe in Section 4.4 (paragraph 3), i.e., self-supervision helps learn a better graph structure compared to if we do not have self-supervision. The reason is that self-supervision allows all edges to receive supervision during training.
> > >
> > > 2.
> > > A more principal explanation has been included in the introduction: “Introducing this self-supervision adds the inductive bias that a graph structure suitable for predicting the node features is also suitable for predicting the node labels.”
> > >
> > > We would like to clarify this with an example. In semi-supervised classification, we would like to predict the labels Y where we only have access to the labels for a small fraction of the nodes. Suppose hypothetically that we have access to some other labels Z for all the nodes such that Z is highly correlated with Y. In this case, if we train our network to not only make good predictions for the observed Y labels but also make good predictions for Z labels (which is available for every node), we can potentially perform better than the case where we only train our model to predict the observed Y labels. That’s because the high correlation between Y and Z enables the model to learn to generalize better to the nodes whose Y value is unobserved (i.e. test nodes).
> > >
> > > Z in the above example corresponds to node features in SLAPS and making predictions for Z corresponds to the self-supervision task.
> > >
> > > Similarly to LDS, in SLAPS, the parameters of the classification GCN model are updated simultaneously with the graph structure (see 4.5). Since this is done through linear interpolation rather than bilevel optimization, it is unclear how the losses interact but we believe that the ablation experiments that we present in the paper show that the denoising task is essential for good performance and that it allows us to learn a graph that is strongly correlated to the classification task.
> > >
> > >
> > > 3.
> > > We intended to propose SLAPS for the case where labeled nodes were available but the underlying graph structure between the nodes was unknown. As suggested by the reviews, we also ran experiments where a noisy graph is known and we have shown that SLAPS can be used successfully there as well. We agree that the work should be compared to models that perform non-graphical classification and we have included the MLP baseline in Table 1 to this effect. We also added the MLP+GAM* baseline suggested by another reviewer. In terms of the few-shot learning methods, they cannot be directly compared to our setting because they focus on learning a representation for new nodes based on a few labels whereas we do not assume that additional labels to the ones seen during training and those used for validation are available.

---

### Decision · Program_Chairs · 2021-01-07
**Final Decision**

**Decision:**

Reject

**Comment:**

The paper received 5 reviews, one of which had positive feedback. Although there are merits associated with the paper, several concerns raised in the reviews and the discussion period that prevents the paper to be accepted. It appears that experiments on noisy graphs are not properly done and competitive baselines are not used for validations. The quality of the learned graph structure is not adequately analyzed. and the experimental setup was not clearly explained. All these indicate that there is a need for a major revision before the paper can be considered for acceptance.